

# SMRT sequencing of the full-length transcriptome of the *Rhynchophorus ferrugineus* (Coleoptera: Curculionidae)

Hongjun Yang[1], Danping Xu[2], Zhihang Zhuo[1,3], Jiameng Hu[1] and Baoqian Lu[4]

[1] Key Laboratory of Genetics and Germplasm Innovation of Tropical Special Forest Trees and Ornamental Plants, Ministry of Education, Key Laboratory of Germplasm Resources Biology of Tropical Special Ornamental Plants of Hainan Province, College of Forestry, Hainan University, Haikou, Hainan, China

[2] Sichuan Provincial Key Laboratory of Agricultural Products Processing and Preservative, College of Food Science, Sichuan Agricultural University, Yaan, Sichuan, China

[3] Key Laboratory of Integrated Pest Management on Crops in South China, Ministry of Agriculture, South China Agricultural University, Guangzhou, Guangdong, China

[4] Key Laboratory of Integrated Pest Management on Tropical Crops, Ministry of Agriculture China, Environment and Plant Protection Institute, Chinese Academy of Tropical Agricultural Sciences, Haikou, Hainan, China

## ABSTRACT

**Background**. Red palm weevil *Rhynchophorus ferrugineus* (Coleoptera: Curculionidae) is one of the most destructive insects for palm trees in the world. However, its genome resources are still in the blank stage, which limits the study of molecular and growth development analysis.

**Methods**. In this study, we used PacBio Iso-Seq and Illumina RNA-seq to first generate transcriptome from three developmental stages of *R. ferrugineus* (pupa, 7th larva, female and male) to increase our understanding of the life cycle and molecular characteristics of *R. ferrugineus*.

**Results**. A total of 63,801 nonredundant full-length transcripts were generated with an average length of 2,964 bp from three developmental stages, including the 7th instar larva, pupa, female adult and male adult. These transcripts showed a high annotation rate in seven public databases, with 54,999 (86.20%) successfully annotated. Meanwhile, 2,184 alternative splicing (AS) events, 2,084 transcription factors (TFs), 66,230 simple sequence repeats (SSR) and 9,618 Long noncoding RNAs (lncRNAs) were identified. In summary, our results provide a new source of full-length transcriptional data and information for the further study of gene expression and genetics in *R. ferrugineus*.

## INTRODUCTION

The red palm weevil (RPW), *Rhynchophorus ferrugineus* (Coleoptera: Curculionidae), is the world's most notorious pest that destroys palm trees (*Vatanparast et al., 2014*; *Wakil, Faleiro & Miller, 2015*). It originated in Southeast Asia and rapidly spread to the rest of the world, including the entire Mediterranean region, Asia and Oceania, destroying the coastal palm trees and threatening the production of coconut trees (*Soroker et al., 2005*). In China, *R. ferrugineus* is an alien invasive pest. Since its invasion, *R. ferrugineus* has led to the death

Corresponding author
Zhihang Zhuo,
zhuozhihang@foxmail.com

of about 20,000 coconut trees, with an area of more than 10,000 square kilometers, which seriously threatens the ecological security of coastal areas (*Shi, Lin & Hou, 2014*; *Ge et al., 2015*). As larva extensively feeds on tissues, the pest would cause extensive damage to the plant's apical meristem, threaten the survival of plants and impair the mechanical stability of plants. When red palm weevil damages on palm trees, the main symptoms are only visible in the later stages of infestation. At this time, the larva has already destroyed the apical meristem, and any control measures are ineffective, eventually causing the infected palm to collapse and dump (*Ferry & Gomez, 2002*; *Sacchetti, 2006*; *Hussain et al., 2013*). The biological properties and continuous feeding behavior of *R. ferrugineus* make it impossible to effectively control this pest by all conventional methods, especially chemical pesticides. Research on biological control strategies, mainly microbes, provides a new approach for controlling *R. ferrugineus*, but their application remains a relatively long-term goal (*Blumberg, 2008*; *Mazza et al., 2014*). Transcriptomes data reflects the information of cellular responses, gene function, evolution and reveal different biological processes at the molecular level (*Hittinger et al., 2010*; *Jia et al., 2018*). Transcriptome analysis improves the understanding of molecular responses and hopefully provide evidence for the genetics and growth of insects, providing references for future studies. For entomology, the research of transcriptome mainly focuses on insect resistance, feeding behavior, mating behavior and oviposition, which will provide new ideas to further study of red palm weevil.

Short-reading transcriptome sequencing has been widely used to describe gene expression levels, and many model organisms and non-model organism transcriptomes were obtained by short-sequence sequencing on a second-generation sequencing platform (*Nagalakshmi et al., 2008*; *Ekblom & Galindo, 2011*; *Djebali et al., 2012*). Recently, the expression analysis of Coleoptera insects at different developmental stages have been studied using the next-generation sequencing method (*Won et al., 2018*; *Chanchay et al., 2019*; *Noriega et al., 2019*). Additionally, the research on transcriptome of *R. ferrugineus* intestinal microbes showed those microbes in the intestine of larva had profound effects on the immune stimulation and nutritional metabolism (*Habineza et al., 2019*; *Muhammad et al., 2019*). However, the spliced transcripts of short-reading transcriptome sequencing are incomplete, and the current short sequence sequencing prediction program is difficult to accurately predict the gene structure (*Coghlan et al., 2008*). Furthermore, low-quality transcripts obtained by short-sequencing sequencing may result in incorrect annotations (*Lin et al., 2017*; *Li et al., 2018*). The second-generation sequencing technology has defect of short read length and can not span the entire transcripts (*Koren et al., 2012*). Nevertheless, third-generation long-read sequencing platforms can overcome those difficulties.

Compared with short-reading transcriptome sequencing, full-length transcriptome sequencing is based on the PacBio Sequel third-generation sequencing platform, which will directly obtain complete transcripts containing 5′UTR, 3′UTR, and polyA tails without interrupting splicing, thereby accurately analyzing reference genomic species. The study on structural information of full-length transcriptome sequencing, such as alternative splicing and fusion genes, solves the problem of short splicing and incomplete information of transcripts without reference genomes. Currently, single molecule real-time long read sequencing (SMRT) is one of the most reliable full-length cDNA molecular sequencing

methods. It has been successfully applied to the full-length transcriptome analysis of human, animals, plants and insects to obtain more authentic transcriptome information reflecting intact species sequence (*Sharon et al., 2013*; *Larsen, Campbell & Yoder, 2014*; *Abdel-Ghany et al., 2016*; *Hartley et al., 2016*; *Wang et al., 2016*; *Chen et al., 2017*; *Zhu et al., 2017*; *Kawamoto et al., 2019*). To the best of our knowledge, there are few reports on full-length transcriptome sequencing of *R. ferrugineus* at present, especially for gene expression analysis at different developmental stages.

In this work, methods of short-reading transcriptome sequencing (Illumina RNA-seq) combined with full-length transcriptome sequencing (PaBio Iso-seq) were applied to obtain a complete full-length transcriptome of *R. ferrugineus* which would be beneficial to comprehensively analyze its transcriptome information. Then, functional annotation, CDS (Coding sequence) prediction, simple sequence repeats analysis, and transcription factors analysis were performed on the complete full-length transcriptome. Finally, lncRNAs and alternative splicing events were analyzed. Here, we performed full transcriptome sequencing for species without a reference genome, providing a complete set of genome reference (transcriptome sequences) of *R. ferrugineus*, supplying a reference for further analysis of gene expression profile, and exhibiting valuable resources for future molecular biology research of red palm weevil.

# MATERIALS & METHODS

## Samples selection and preparation

All *R. ferrugineus* samples used in this study were collected from the Coconut Research Institute, Chinese Academy of Tropical Agricultural Sciences, Wenchang Hainan, China. Samples were divided into larva, pupa, female adult and male adult. The whole body was collected from 12 *R. ferrugineuss* (3 from 7th instar larvae, 3 from Pupae, 3 from female adults, 3 form male adults). All samples were harvested and frozen in liquid nitrogen and stored at −80°C for further experiments.

## RNA extraction and SMRT sequencing

Total RNA samples were isolated using the RNeasy Plus Mini Kit (Qiagen, Valencia, CA, USA). Then 1% agarose gels was used to detect RNA degradation and contamination, and Nanodrop (NanoDrop products, USA) was used to check RNA purity (OD 260/280). RNA concentration and integrity were accurately evaluated using Qubit® RNA Assay Kit in Qubit® 2.0 Flurometer (Life Technologies, CA, USA) and Agilent 2100 (Agilent Technologies, USA), respectively. For PacBio Iso-Seq, the total RNA samples from three developmental stages (larva, Pupa, female adult and male adult) were mixed together for the following experiments. A total of 3 μg of mixed RNA was sequenced on the Pacbio Sequel platform (Pacific Biosciences, CA, USA) in accordance with the manufacturer's instructions. Then, according to the Isoform sequencing protocol (Iso-Seq), the Iso-Seq library was prepared by the Clontech SMARTer PCR cDNA synthesis kit (Clontech, CA, USA) and the BluePippin size selection system protocol described by Pacific Biosciences (PN 100-092-800-03). For Illumina RNA-Seq, twelve libraries of three developmental stages (larva, pupa, male adult and female adult) RNA samples were prepared and

sequenced respectively. For each sample, a total amount of 1.5 μg RNA was used for short reads sequencing on Novaseq 6000 platform. Sequencing libraries were generated using NEBNext® Ultra™ RNA Library Prep Kit for Illumina® (NEB, USA) following manufacturer's recommendations and index codes were added to attribute sequences to each sample. The sequencing work reported in this work was performed by Novogene technology co. (Beijing, China). PacBio Iso-Seq and Illumina RNA-seq data generated from *R. ferrugineus* are available from the NCBI SRA database under project number PRJNA598560.

## Data processing and error correction of PacBio Iso-Seq reads

Firstly, SMRTlink 6.0 software was used to process the sequence data. Immediately after, the cyclic consensus sequence (CCS) was generated from the subread BAM files (parameters: min_length 50, max_drop_fraction 0.8, no_polish TRUE, min_zscore-9999.0, min_passes2, min_predicted_accuracy 0.8, max_length 15000) and a CCS.BAM file was output. The generated BAM files were divided into full-length and non-full-length reads using pbclassify. Finally, input the full and non-full length fasta files into the clustering step, which performs the isoform level clustering, and then uses Quiver (parameters: hq_quiver_min_accuracy 0.99, bin_by_primer false, bin_size_kb 1, qv_trim_5p 100, qv_trim_3p 30) for the final arrow polishing. Full-length transcripts were corrected using Illumina RNA-seq data with the software LoRDEC (*Salmela & Rivals, 2014*). The redundancies in the corrected transcript were then removed using the CD-HIT-EST (parameters: -c 0.95 -T 6 -G 0 -aL 0.00 -aS 0.99) program to obtain the final transcript for subsequent analysis (*Fu et al., 2012*).

## Functional annotation of transcripts

Transcripts function was annotated based on the following databases:NR (NCBI non-redundant protein sequences) (*Deng et al., 2006*), NT (NCBI non-redundant nucleotide sequences), Pfam (Protein family) (*Finn et al., 2014*), KOG (Clusters of Orthologous Groups of proteins) (*Koonin et al., 2004*), Swiss-Prot (A manually annotated and reviewed protein sequence database) (*Apweiler et al., 2004*), KEGG (Kyoto Encyclopedia of Genes and Genomes) (*Kanehisa et al., 2004*) and GO (Gene Ontology) (*Ashburner et al., 2000*). The BLAST software with $E$-value $\leq 1e-5$ was used for NT database analysis. The Diamond v0.8.36 software with $E$-value $\leq 1e-5$ was analyzed in NR, KOG, Swiss-Prot and KEGG annotations. The Hmmscan procedure was used in the Pfam database, and GO function categories were performed by Blast2GO v2.5 based on Pfam annotation.

## CDS prediction and SSR analysis

The ANGEL pipeline is a long-read implementation of ANGLE that is used to determine protein coding sequences from cDNAs. We use the confidence protein sequences of *R. ferrugineus* or closely related species for ANGLE training, and then run the ANGLE predictions for given sequences (*Shimizu, Adachi & Muraoka, 2006*). At the same time, the MISR (http://pgrc.ipk-gatersleben.de/misa/misa.html) was used to identify and locate the SSR of the transcriptome.

## Identification of TFs and lncRNAs

Animal transcription factors were performed by the animal TFDB 2.0 database (*Zhang et al., 2015*). For species included in the database, if they were not Ensembl geneid genes, they would be screened by BLASTX with the known transcription factors protein sequence of the species in the database, and if they were Ensembl geneid, they would be screened directly. For species not included in the database, hmmsearch was used to identify them according to pfam files of the transcription factor family. LncRNA of the transcriptome were predicted by using Coding-Non-Coding-Index (CNCI) (*Altschul et al., 1997*), Coding Potential Calculator (CPC) (*Kong et al., 2007*), Pfam-scan (*Finn et al., 2016*) and PLEK (*Li, Zhang & Zhou, 2014*) to predict the coding potential of transcripts. Firstly, PLEK SVM classifier with default parameters of—minlength 200 and CNCI with default settings were performed to evaluate coding potential; Secondly, CPC and NCBI eukaryotic protein database were used for BLAST comparison (*E*-value < 1e−10 setting) to search transcripts; Finally, homologous search of hmmscan was performed for the transcription sequences predicted by the three software with the Pfam database, and the protein family domains were recorded, with the default parameter of -E 0.001-domE 0.001.

## AS analysis

To obtain alternative splicing (AS) events for *R. ferrugineus*, transcripts were further processed using Coding GENome reconstruction tool (Cogent v3.1, https://github.com/Magdoll/Cogent). Generally, Cogent first divides the input fasta file into chunk_size blocks, and then calculates the k-mer configuration file. Then, the De Bruijn graph was used to further reconstruct each transcription family into one or more unique transcription models (called UniTransModels). Finally, gmap-2017-06-20 was conducted to map the adjusted transcripts to UniTransModels. Splicing junctions detection was performed on transcripts mapped to the same UniTransModels, and transcripts with the same splice junctions were collapsed. Meanwhile, the transcriptional isoforms of UniTransModels have collapsed transcripts with different splicing junctions. AS events were detected with SUPPA (https://github.com/comprna/SUPPA) using default settings.

# RESULTS

## The full-length sequences of *R. ferrugineus* using PacBio sequencing

The full-length transcriptome of *R. ferrugineus* was generated using the PacBio Sequel platform on the pooled RNA of twelve *R. ferrugineus* samples. The results showed that PacBio Sequel platform generated a total of 454,369 circular consensus sequences (CCSs) with a full length reads of 362,466. The nonfull-length (nFL) sequences was 81,424 and the full-length non chimera (FLNC) reads number was 330,973 with an average length of 2,332 bp. Twelve samples were sequenced by Illumina Novaseq 6000 respectively, and a total of 642,179,304 raw reads and 625,983,256 clean reads (97.48%, 93.91 G) were obtained. PacBio Sequel platform produced a total of 10,172,136 subreads and 181,405 consensus reads (16.67 G, with an average length of 2,305 bp, an N90 of 1,327 bp and an N50 of 2,790 bp), which were then corrected using the Illumina reads (after correction, the average

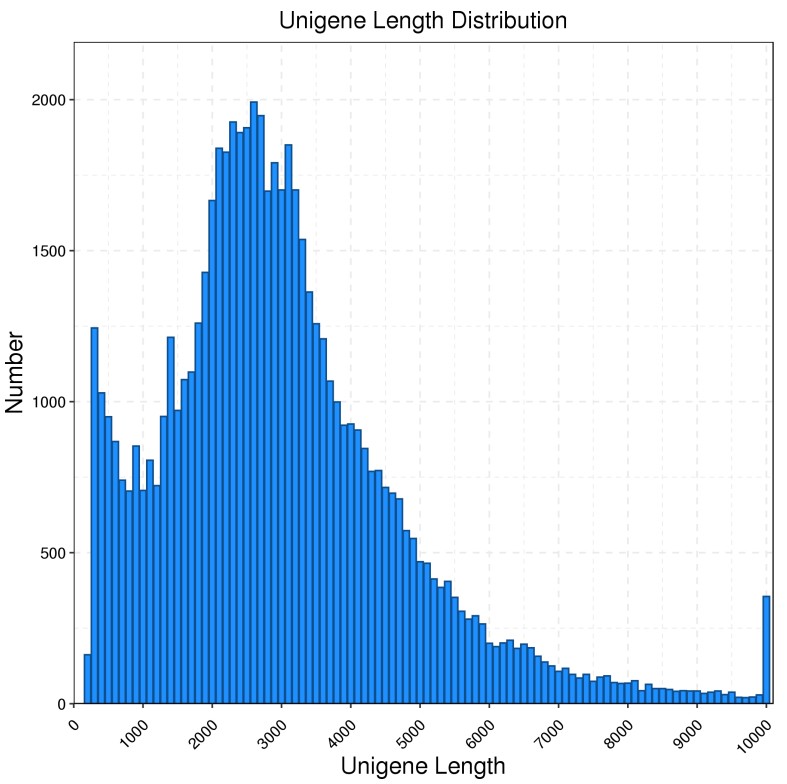

**Figure 1  Length distribution of *R. ferrugineus* unigenes obtained by PacBio Iso-Seq.**

length was 2,302 bp, N90 was 1,321 bp, and N50 was 2,785 bp) and ubsequently removing redundancy via the CD-Hit program, the consensus transcripts were finally clustered into a total of 63,801 transcripts for subsequent analysis. We found that the main length distribution range of unigenes was 0.5–6 k (Fig. 1).

## Gene annotation of *R. ferrugineus*

To obtain a comprehensive functional annotation of *R. ferrugineus* transcriptome, we annotated 63,801 transcripts with seven databases, including Swiss-Prot, KOG, GO, NR, NT, Pfam, and KEGG. In total, 50,280, 40,109, 47,197, 33,511, 27,707, 27,253 and 27,707 transcripts were annotated in the NR, Swiss-Prot, KEGG, KOG, GO, NT and Pfam databases, respectively. Moreover, 54,999 transcripts were annotated in at least one database and 12,508 transcripts were annotated in all databases (Fig. 2).

NR is a non-redundant protein database characterized by its comprehensive content and the inclusion of species information in the annotated results, which can be used for the classification of homologous species. Aligned all transcripts in the NR database, the results showed that 50,280 transcripts were annotated in the NR database and the top five most annotated in NR database were *Dendroctonus ponderosae, Anoplophora glabripennis, Bactrocera tryoni, Tribolium castaneum* and *Aethina tumida* (Fig. 3).

KOG is a protein database created and maintained by NCBI, which is constructed according to the phylogenetic relationship of coding proteins in complete genomes of

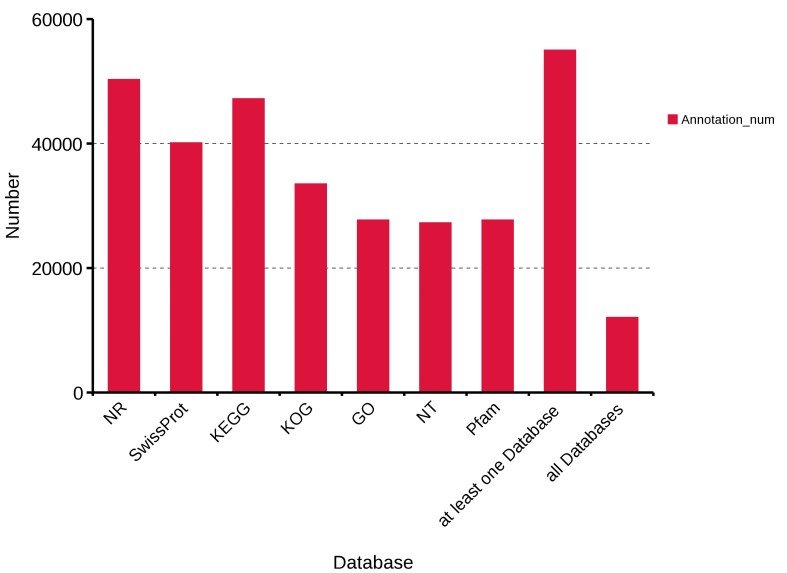

**Figure 2** **Function annotation of *R. ferrugineus* transcripts in all databases.** NR, Non-Redundant Protein Database. Swiss-Prot, a manually annotated and reviewed protein sequence database. KEGG, Kyoto Encyclopedia of Genes and Genomes. KOG, euKaryotic Ortholog Groups. GO, Gene Ontology. NT, NCBI non-redundant nucleotide sequences. Pfam, Protein family.

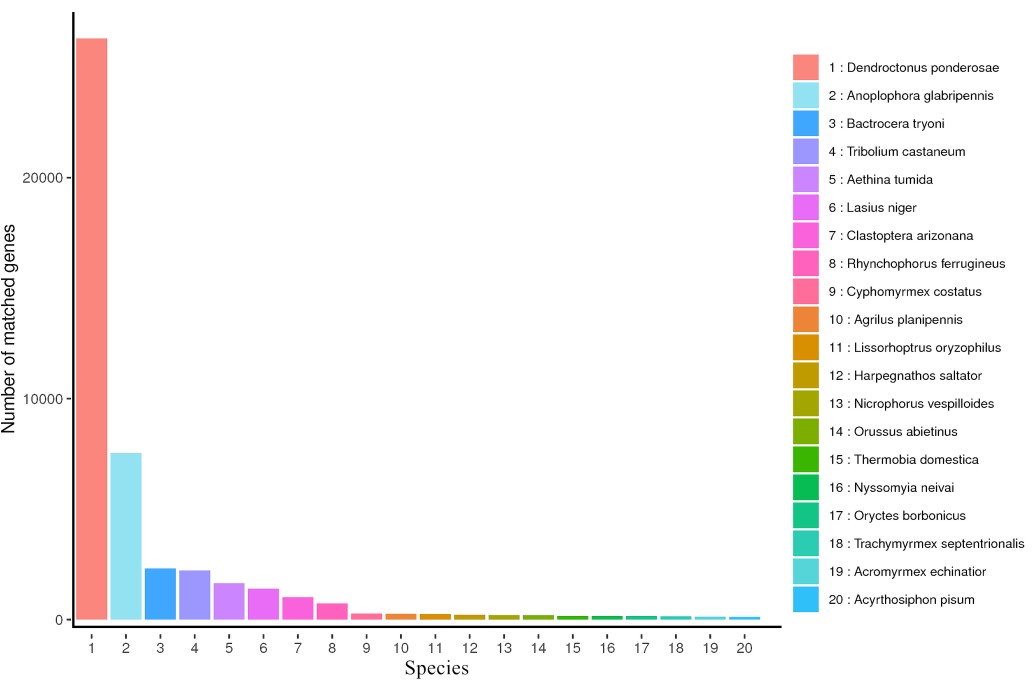

**Figure 3** **NR Homologous species distribution diagram of *R. ferrugineus* transcripts.**

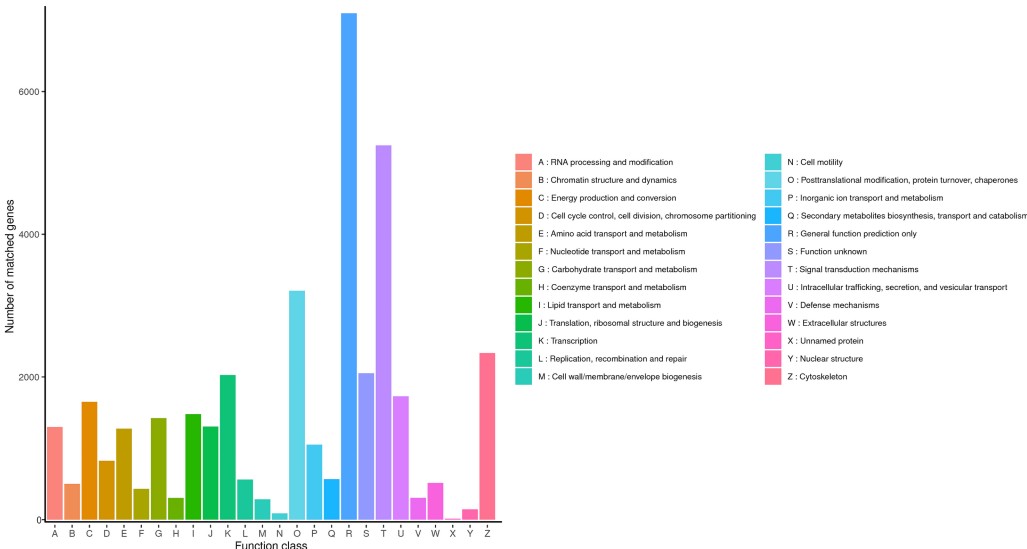

**Figure 4**   **KOG classification diagram of *R. ferrugineus* transcripts.**

bacteria, algae and eukaryotes, and is widely used to predict the function of sequences. The KOG functional classification of *R. ferrugineus* transcripts was shown in Fig. 4. The results indicated that a total of 33,511 genes categorizing into 26 categories were annotated in KOG database. The first six largest groups among these categories were transcription, general function prediction only, function unknown, signal transduction mechanisms, cytoskeleton and posttranslational modification (protein turnover and chaperones), respectively.

Full-length transcripts of red palm weevil were annotated with GO database, and 27,707 annotated transcripts were successfully divided into three categories: biological processes, molecular functions, and cellular Components (Fig. 5). In biological process, the cell process accounts for the largest proportion, followed by metabolic process and single-organism process. In addition, we also found that part of the genes was annotated into biological regulation, regulation of biological process, localization, response to stimulus and signaling terms. In cellular component, the genes involved in cell, cell part, organelle, membrane, membrane part and macromolecular complex were the most. In molecular function categories, binding, catalytic activity and transporter activity were identified as the most abundant terms.

KEGG Pathway analysis can be used to systematically analyze the metabolic pathways of gene products and compounds in cells and the functions of these gene products. In the KEGG classification of *R. ferrugineus*, human diseases, metabolism and organismal systems were the top three categories with higher proportions (Fig. 6). Briefly, a total of 11,950 genes were involved in human disease related pathways, in which 1,668 genes were predicted to infectious disease: viral, 1,396 genes were predicted tonfectious diseases: bacterial and 2,777 genes were predicted to cancers: overview. A total of 10,294 of the annotated genes were classified as belonging to organismal systems related pathways, in which the nervous system (1,016 genes), immune system (1,984 genes), digestive system (993 genes) and

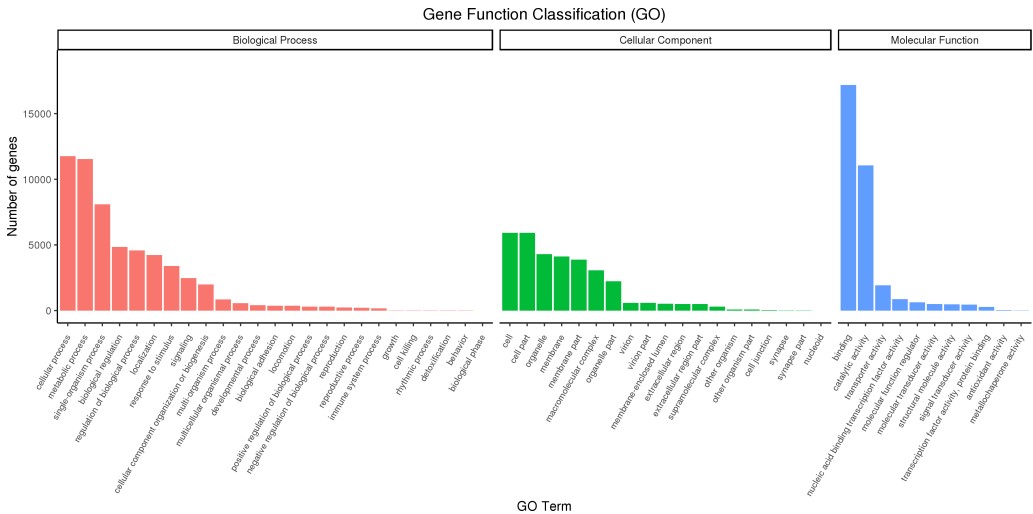

**Figure 5** GO classification diagram of *R. ferrugineus* transcripts.

endocrine system (2,500 genes) were the top four pathways with the most abundant genes. In addition, a total of 7,979 annotated genes were involved in the Metabolism pathway. The most abundant pathways were carbohydrate metabolism (1,478 genes) and amino acid metabolism (1,180 genes). Regarding Environmental Information processing, genes were involved in the signal transduction (4,741 genes), signaling molecules and interaction (304 genes) and membrane transport (196 genes). At the same time, a lower number of genes are annotated to Cellular Processes and Genetic Information Processing.

## CDS prediction

The CDS is a sequence encoding a protein product that corresponds exactly to the codon of a protein. In the sequencing results of the full-length transcriptome, predicting the protein coding region contributes to the preliminary analysis of the gene and is also the basis for subsequent protein structure analysis. For red palm weevil, ANGEL software was performed to carry out CDS prediction analysis on the obtained full-length transcriptome sequence, and the results showed that the main distribution range of CDS length was 0 ~2,500 nt (Fig. 7).

## Transcription factors identification

Transcription factors are an important part of the transcriptional regulatory system. Using the present data of *R. ferrugineus*, 2,084 transcription factors were predicted, and Zf-C2H2 (570,27.35%), ZBTB (476,22.84%), TF_bzip (101,4.85%) and bHLH (85, 4.08%) were the top four transcription factor families (Fig. 8). These transcription factors will lay the foundation for exploring the role of the regulatory mechanism of red palm weevil.

## SSR discovery

Simple sequence repeats is a group of repeated sequences consisting of several nucleotides (1~6) with repeat units up to dozens of nucleotides. The repeats are short in length

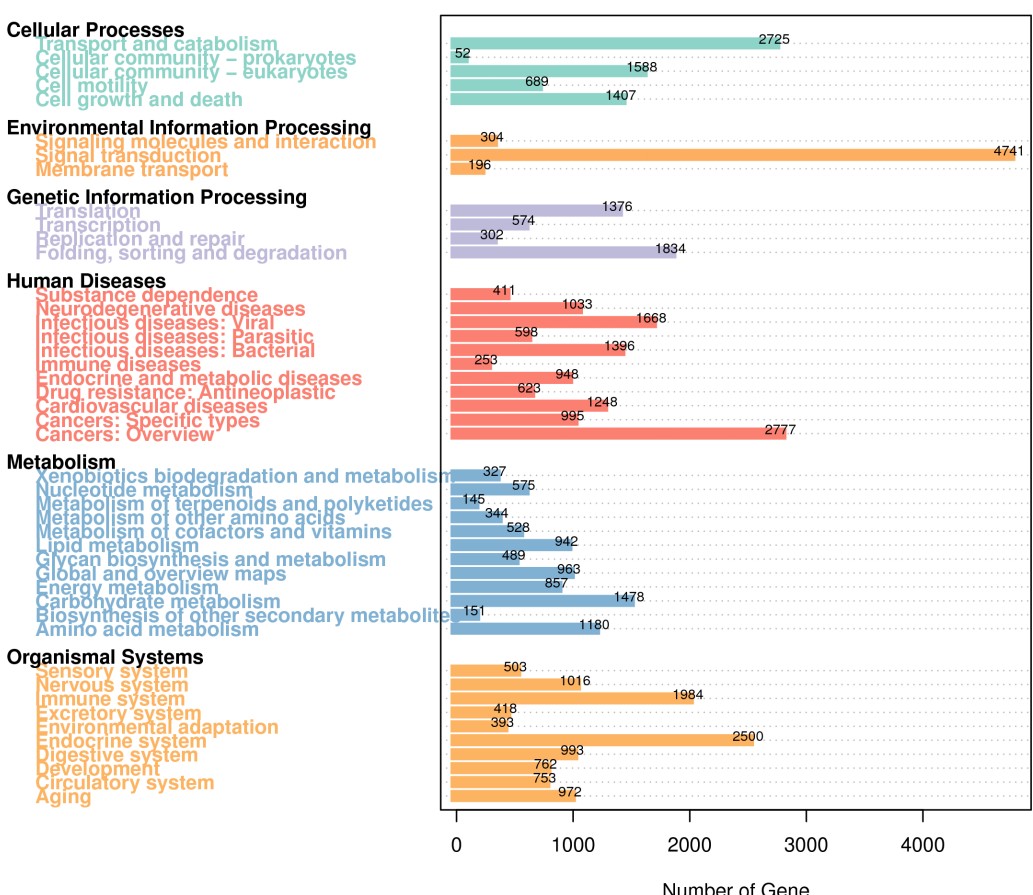

**KEGG pathway annotation**

**Figure 6  KEGG Pathway classification diagram of *R. ferrugineus* transcripts.**

and widely distributed uniformly in eukaryotic genomes. In our analysis, MISA software (version 1.0, default parameter) was applied for SSR detection of transcriptome. The results showed that the minimum number of repetitions of each unit size is 1–10, 2–6, 3–5, 4–5, 5–5, 6–5. In total 66,230 SSR loci were identified in this transcriptome; mono nucleotide motifs (49,898, 75.34%) were the most abundant type of SSR locus, followed by di nucleotide motifs (12,662, 19.12%), tri nucleotides(3,377, 5.09%), tetra nucleotides (192,0.29%), penta nucleotides (33, 0.05%) and hexa-nucleotide motifs (68, 0.10%) (Fig. 9).

## LncRNA prediction

LncRNA (long-chain noncoding RNA) is a class of RNA molecules whose transcripts are more than 200 nt in length and do not encode proteins. For *R. ferrugineus*, the numbers of lncRNAs identified from transcriptome by CNCI, CPC, PLEK and Pfam were 32,552, 17,481, 18,897 and 34,066, respectively (Fig. 10). The intersection of these four results

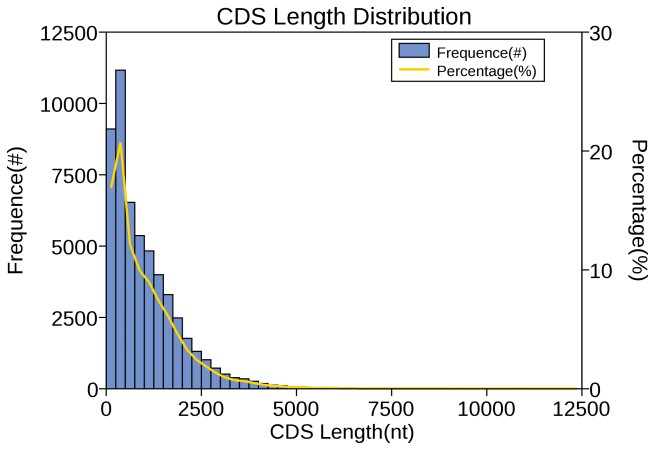

**Figure 7** Number, percentage and length distributions of coding sequences of *R. ferrugineus* transcripts.

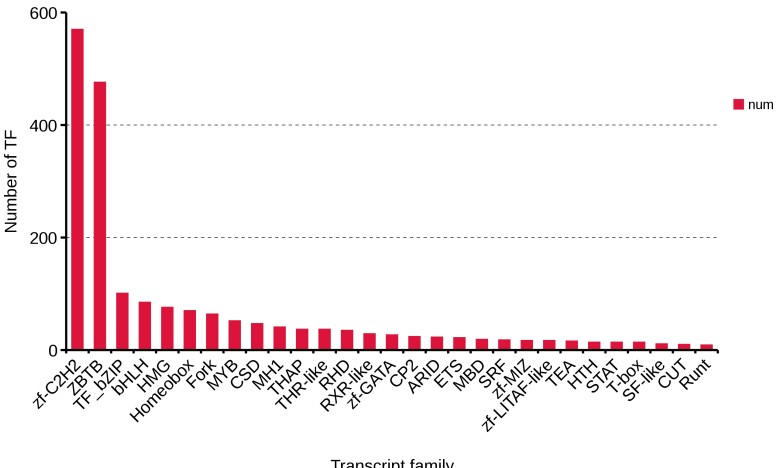

**Figure 8** Number and family of top 29 transcription factors predicted by SMRT.

produced 9,618 lncRNA transcripts. Meanwhile, the length distribution density of mRNA was compared with the predicted lncRNA (Fig. 11).

## Alternative splicing analysis from full-length transcriptomes

Since *R. ferrugineus* had no reference genome, we used Cogent (Coding genome reconstruction tool) to reconstruct genes using high-quality full-length transcriptome sequences to generate UniTransModels. UniTransModels were used as a reference sequence to describe the types of AS events and the number of corresponding genes. The results indicated that a total of 2,184 UniTransModels-based AS events in *R. ferrugineus* were detected. Briefly, six main AS events (alternative 3′ splice sites, Mutually exclusive exons, Skipping Exon, alternative 5′ splice sites, Retained introns and Alternative First Exons) were identified. Retained introns (RI) were identified as the most abundant event, accounting for

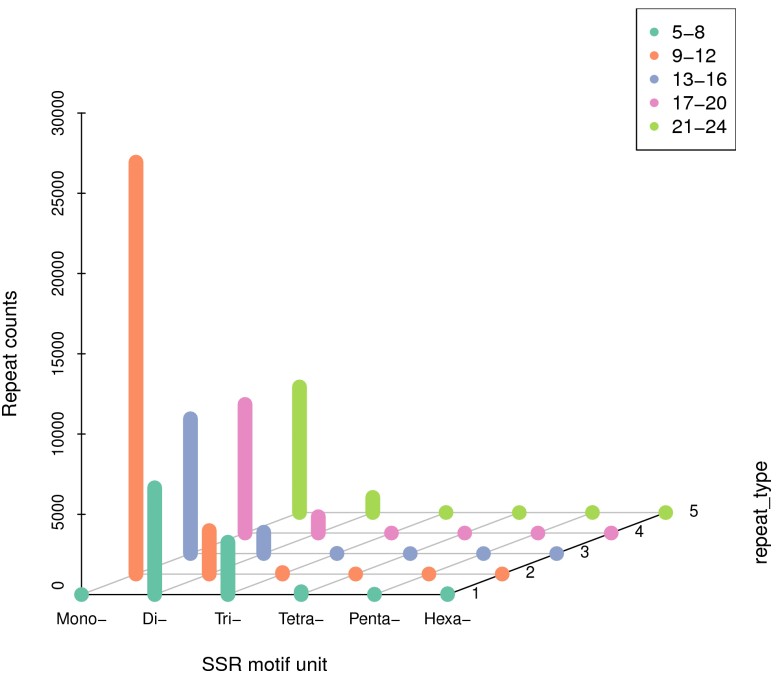

**Figure 9** **Scattergram of simple sequence repeats of *R. ferrugineus* transcripts.**

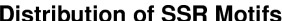

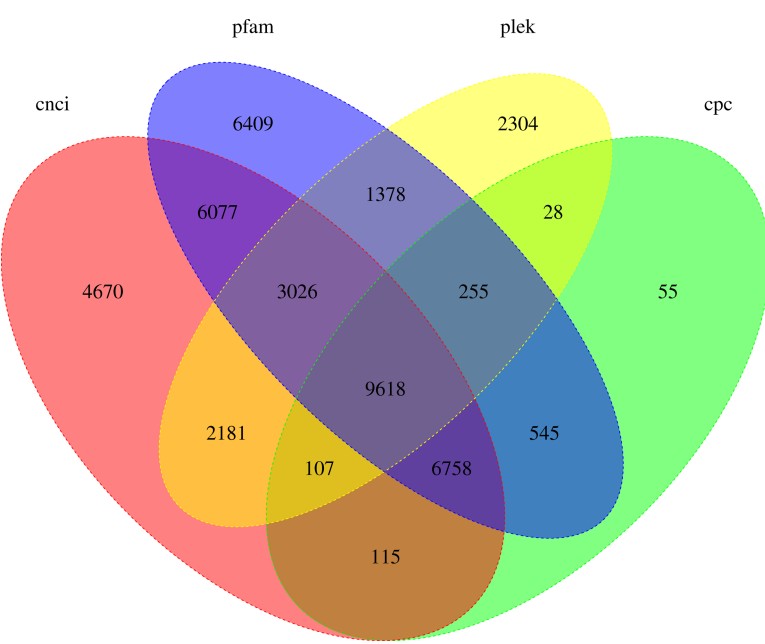

**Figure 10** **Venn diagram of lncRNA transcripts identified from PLEK, CNCI, CPC and Pfam.**

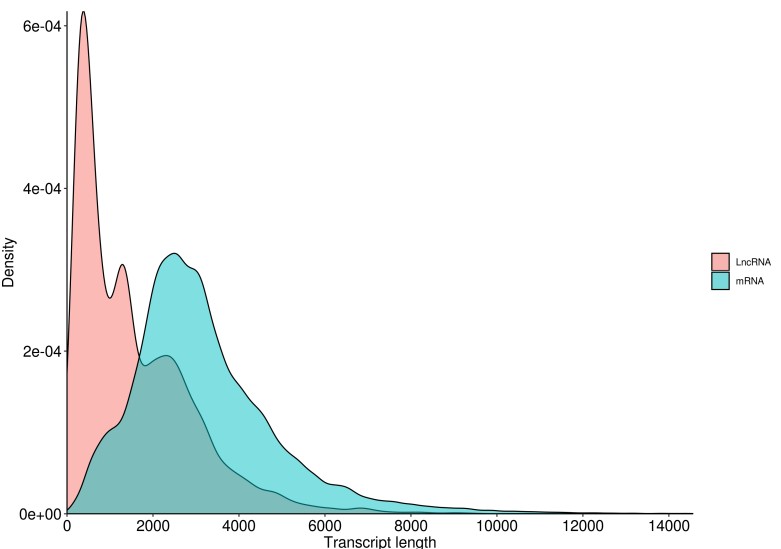

**Figure 11  Length distribution of LncRNA and mRNA in *R. ferrugineus.***

6.14% (134) of all events. The other five types of AS events account for less than 2% of all detectable events. The number of two kinds of events, alternative 3′ splice sites (32,1.47%) and alternative 5′ splice sites (34,1.56%), were slightly higher than those of skipped exons (7,0.32%), mutually exclusive exons (1,0.05%) and Alternative first exons (20,0.92%).

## DISCUSSION

In recent years, with the development of sequencing technology, transcriptome sequencing has become an important mean to study the regulation of gene expression. The third-generation sequencing (full-length transcriptome sequencing) captures full-length transcripts without assembly, overcoming the difficulties of the second-generation sequencing (short-reading transcriptome sequencing). Besides, the third-generation sequencing contributes to the following fields: accurately reflect the transcriptome information of the sequenced species; detect multiple variable splicing forms, and find more splicing sites and alternative splicing events; find new functional genes, and supplement the genome annotation; accurately analyze fusion genes, homologous genes, superfamily genes and alleles. According to the report, for *Oxya chinensis*, *Acrida cinerea* and *Atractomorpha sinensis*, the total length, average length, N50 and N90 of transcripts obtained by PacBio RS II platform (full-length transcriptome) all greater than those obtained by RNA-seq transcriptomes (short-reading transcriptome) (*Zhao, 2018*). Although the third-generation sequencing technology represented by PacBio Iso-Seq has the superiority of extremely long reading length, its reading error rate of single-base is pretty high (up to 15%), which can be corrected by second-generation short reads (*Au et al., 2013*; *Li et al., 2014*). In this work, we pooled and sequenced RNA samples from different developmental stages of *R. ferrugineus*, using both PacBio Iso-Seq and Illumina RNA-seq to obtain full-length transcriptome data, providing a general encyclopedia of gene transcription. As expected, massive transcriptome

data of *R. ferrugineus* was generated, including 63,801 full-length transcripts with 2,964 bp of average length and 3,547 bp of N50 length. The amount of transcriptome data acquired in red palm weevil was much higher than that in Coleoptera insects with different developmental stages by second-generation sequencing, such as *Hypothenemus hampei* (average length 1,609.92 bp and N50 length 2,427 bp) (*Noriega et al., 2019*), *Nicrophorus orbicollis* (average length 1,193 bp and N50 length 2,856 bp) (*Won et al., 2018*), *Sclerotia aquatilis* (average length 1,394 bp and N50 length 2,666 bp) (*Chanchay et al., 2019*), and *Batocera horsfieldi* (average length 1,188 bp and N50 length 3,143 bp) (*Yang et al., 2018*). Meanwhile, a total of 54,999 (86.20%) among 63,801 transcripts of *R. ferrugineus* were successfully annotated as known homologous genes using seven databases.

In the study of gene annotation, a large number of new transcripts can be classified for obtaining gene function information. The updated collection of homologous protein sets of prokaryotes and eukaryotes is expected to be used for functional annotation of newly sequenced genomes, including those complex eukaryotes, as well as genome-wide evolutionary studies (*Tatusov et al., 2003*). Genomic sequencing has made it clear that a large fraction of the genes specifying the core biological functions are shared by all eukaryotes (*Ashburner et al., 2000*). In SMRT sequencing of the full-length transcriptome of the *R. ferrugineus*, a total of 27,707 FL transcripts were annotated into the GO database, most of which were biological processes, followed by cellular components and molecular functions. A total of 47,197 transcripts of red palm weevil were annotated to 40 KEGG pathways, the top four most annotated KEGG pathways were signal transduction, cancers (specific types), transport and catabolism, endocrine system. In addition, the KOG annotation results showed that the transcripts associated with General function prediction only and Signal transduction mechanisms were the most. The results of gene annotation indicated that the new transcripts were related to the above functions.

AS events have attracted the attention of biologists as an important mechanism to increase protein diversity and regulate gene expression (*Thatcher et al., 2016*; *Vuong, Black & Zheng, 2016*). AS occurs by rearranging the pattern of intron and exon elements that are joined by splicing to alter the mRNA coding sequence (*Braun et al., 2018*). PacBio long-read transcriptome sequencing is superior to short-read RNA-Seq in the recognition of AS events (*Tilgner et al., 2014*; *Weirather et al., 2015*). At the same time, the accuracy of the PacBio transcript to identify AS events has been verified (*Abdel-Ghany et al., 2016*; *Wang et al., 2016*). In our work, third-generation sequencing technology was adopted to capture 2,184 AS events from the FL transcripts.

lncRNAs mainly regulate gene expression at the epigenetic level through transcriptional regulation and post-transcriptional regulation, and exert powerful biological functions by affecting protein localization and telomere replication (*Batista & Chang, 2013*; *Kung, Colognori & Lee, 2013*; *Qureshi & Mehler, 2013*). In recent years, a large number of lncRNAs have been identified from insects, such as *Apis mellifera*, *Nasonia vitripennis* and *Nilaparvata lugens*, which laid an important foundation for further study of the function of lncRNAs in insect growth and development (*Zhu, Liang & Gao, 2016*). Furthermore, the functions of lncRNAs in *Drosophila* have been extensively studied in multiple insect species. For example, lncRNA can be involved in regulating the sex determination process

of *Drosophila* (*Mulvey et al., 2014*), male courtship behavior (*Chen et al., 2011*), motor behavior and climbing ability (*Li et al., 2012*), inactivation of X chromosome (*Smith, Allis & Lucchesi, 2001*), and sleep behavior (*Soshnev et al., 2011*). In *Plutella xylostella*, the lncRNA regulates the resistance of the insect to bacillus thuringiensis (bt) endotoxin Cry1Ac, phenylpyrazole and chlorpyrifos (*Etebari, Furlong & Asgari, 2015*). In addition to regulating growth and drug resistance, lncRNAs have a rapid response to stress and stimulation (*Lakhotia, 2012*; *Mizutani et al., 2012*; *Valluri, Rupam & Srividya, 2017*; *Li et al., 2019*). Many studies have confirmed that lncRNAs can modulate multiple immune responses, including several pathways related to innate immunity (*Fitzgerald & Caffrey, 2014*; *Heward & Lindsay, 2014*; *Ahmed & Liu, 2018*). Simultaneously, some transcription factors may be involved in different metabolic processes and may have multiple different functions (*Chen & Rajewsky, 2007*). For example, GATA and FoxA transcription factors play an important role in the differentiation and maintenance of different tissues by controlling gene expression (*Boyle & Seaver, 2010*; *Zaret & Carroll, 2011*). In some insects, AhR/ARNT may regulate the overexpression of multiple detoxification genes related to pesticide resistance (*Hu et al., 2019*). Transcription factor *limpet* has an impact on fungus-free insect survival, and these transcription factors have a direct effect on protecting *Triatoma infestans* from conditions of pathogenic pathogens, and these transcription factors are part of the primary immune response of other insects (*Altincicek, Knorr & Vilcinskas, 2008*; *Jin et al., 2008*; *Mannino, Paixão & Pedrini, 2019*). A total of 9,618 lncRNAs and 2,084 transcription factors of *R. ferrugineus* were obtained in our study. Furthermore, the discovery of these lncRNAs and TFs will provide certain reference information for further research on the function of *R. ferrugineus* in the growth, immunity and insecticide resistance.

## CONCLUSIONS

PacBio Iso-Seq and Illumina RNA-seq were combined to successfully perform a full-length transcriptome of *R. ferrugineus*, and analyses of gene annotation, CDS prediction, transcription factors, SSR discovery, LncRNA prediction and alternative splicing were smoothly conducted without reference genome species. This research provides a valuable set of complete full-length transcripts for genomic reference, supplying an important and valuable basis for further study of the growth and development of *R. ferrugineus*, as well as other congeneric insects.

**Abbreviations**

| | |
|---|---|
| **FL** | full-length |
| **SMRT** | single-molecule real-time |
| **AS** | alternative splicing |
| **TFs** | transcription factors |
| **SSR** | simple sequence repeats |
| **lncRNAs** | long noncoding RNAs |
| **CDS** | coding sequence |
| **CCSs** | circular consensus sequences |

| nFL | nonfull-length |
|---|---|
| FLNC | full-length non chimera |
| NR | Non-Redundant Protein Database |
| Swiss-Prot | A manually annotated and reviewed protein sequence database |
| Nt | NCBI non-redundant nucleotide sequences |
| Pfam | Protein family |
| GO | Gene Ontology |
| KOG | euKaryotic Ortholog Groups |
| KEGG | Kyoto Encyclopedia of Genes and Genomes |
| CNCI | Coding-Non-Coding-Index |
| CPC | Coding Potential Calculator |
| F_Adult | female adult |
| M_Adult | male adult |
| larva | 7th instars larva |

## ACKNOWLEDGEMENTS

We thank Professor Wei Yang for technical support (Provincial Key Laboratory of Forest Protection, College of Forestry, Sichuan Agricultural University, Yaan City, China). We thank Professor Wei Yan for samples support (Coconut Research Institute, Chinese Academy of Tropical Agricultural Sciences, Wenchang City, China). We also wish to express our deep appreciation to the anonymous reviewers for their comments that helped to improve the quality of this manuscript.

### Funding

This work was supported by the Hainan Natural Science Foundation Project (NO. 319QN163), The Fund Project of Key Laboratory of Integrated Pest Management on Crops in South China, Ministry of Agriculture, P. R. China (NO. SCIPM2018-04) and the Research Initiation Project of Hainan University (NO. KYQD[ZR]1823). The funders had no role in study design, data collection and analysis, decision to publish, or preparation of the manuscript.

### Grant Disclosures

The following grant information was disclosed by the authors:
Hainan Natural Science Foundation Project: 319QN163.
The Fund Project of Key Laboratory of Integrated Pest Management on Crops in South China, Ministry of Agriculture, P. R. China: SCIPM2018-04.
Research Initiation Project of Hainan University: KYQD[ZR]1823.

### Competing Interests

The authors declare there are no competing interests.

## Author Contributions

- Hongjun Yang and Zhihang Zhuo conceived and designed the experiments, performed the experiments, analyzed the data, prepared figures and/or tables, authored or reviewed drafts of the paper, and approved the final draft.
- Danping Xu analyzed the data, prepared figures and/or tables, and approved the final draft.
- Jiameng Hu performed the experiments, prepared figures and/or tables, and approved the final draft.
- Baoqian Lu conceived and designed the experiments, authored or reviewed drafts of the paper, and approved the final draft.

## Data Availability

SMTR sequencing and Illumina RNA-seq data generated from Rhynchophorus ferrugineus are available at NCBI SRA: PRJNA598560.

## Supplemental Information

Supplemental information for this article can be found online at http://dx.doi.org/10.7717/peerj.9133#supplemental-information.

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
