# Peer review of "SMRT sequencing of the full-length transcriptome of the Rhynchophorus ferrugineus (Coleoptera: Curculionidae)"

_PeerJ, doi:10.7717/peerj.9133_

## Round 0.1 · original submission · Major Revisions

Dear Dr. Zhuo,

Your MS describes relevant data for the field and the acquisition of these data followed consistent methods. The MS has no major flaw.

However, as noted by both reviewers, the MS needs substantial improvement before it can be accepted for publication. Please read carefully their detailed comments bellow. As a general rule, I emphasize the need of a more complete and detailed methods section, and the description in the main MS text of where the raw data were deposited, including the accession numbers.

I also agree with the reviewers that the Introduction and Discussion could be improved. A comparison with other Coleoptera transcriptome and a brief discussion on how this data would assist pest control policies would greatly benefit the MS.

Please, also consider to review the figures and its legends.

Kind regards,

Reviewer 1 ·

Basic reporting

The article could use some editing, some sentences need to be improved for clarity, such as lines 157-158, and also some vague, yet over-promissing sentences (e.g. 77-79) need rephrasing. The English reads clearly as non-native, but is serviceable with minor editing.

Experimental design

The Methods section need major changes, and some parts of the methodology is not clearly written or just absent. This is a major issue for any attempt of reproducing the results, and also does not clearly describes what was done.

Beginning at line 123, the sequencing section of the methodology describes how the authors extracted the DNA and the PacBio sequencing, but it should also include which sequencing kit was utilized, and which generation of PacBio machine was used. It matters as it informs what fragment size could have been obtained, and the amount of data generated. The authors mention more than once that they used Illumina sequencing technology to check for sequencing errors, and it is a good and safe thing to do. But the authors never mention how much data they generated for Illumina reads, what sequencing machine and kit were used or any details whatsoever. This is a major flaw that needs correction. I could find some of this information by accessing the raw data NCBI link, but it needs to be on the manuscript, specially since not all details are sent to NCBI.

Some other methods were very vague and unclear. The “Functional annotation of transcripts” (line 149), and “Identification of TFs and lncRNAs” (line 162) only list the databases used in these steps, but fail to mention what exactly was done. The functional annotation, for example, only describes all major databases and vaguely mention they were used (line 150) to “annotate” the transcripts function, but never tells the method, programs, and parameters used. This is not only bad for the reader if they want to reproduce such steps, but also completely non-descriptive of whatever was done. I believe this is unacceptable in a manuscript.

The CDS prediction (lines 157-159) are confusing and make no sense. They should be rewritten.

On line 171 (AS analysis) the authors mention “De Bruijn diagram” instead of “De Bruijn graph”, which is mathematically not the same thing and should be corrected.

In the Results section (line 187) the authors mention again the Illumina data, and point to Table 1, which is very short and not really informative. Table 1 should either be expanded or just written down in text form since it’s two lines only. It also contain a typo.

Validity of the findings

This is a very straightforward manuscript that offers to the community new transcriptome data for a species that doesn’t have an assembled genome yet. While it is a description paper, I would like to see certain results further discussed in the Discussion section. For example, the LncRNA prediction has its amount described, but not really explored or explained in the Discussion.

I understand this is mostly a preliminary report, or a descriptive paper, but I would like a further exploration of the data acquired by the authors, perhaps a comparison with other Coleoptera transcriptome data currently available could enrich the discussion section.

Additional comments

This is a short, descriptive, paper that provides the community with a new, useful, dataset. I appreciate the effort into publishing a manuscript with mostly new data that will serve the scientific community. I just wish the authors would put a little extra effort into making the methods more clear and detailed for the reader and perhaps discuss the findings in light of other available datasets.

Reviewer 2 ·

Basic reporting

The article should be rewriten. Several language mistakes, lack of proper punctuation, even some sentences starting with a nonsense word in lowercase.
Weak reference library.
The author was more committed to reinforce the advantages of third generation sequencing than exploring his hypothesis. In fact, the hypothesis only appears in the introduction.
The figures and tables legends were poorly written.
They should compare and discuss about another Colleoptera transcriptomes, argue and convince the reader that his hypothesis was corroborated by the results.
There is no Discussion. The author repeated some results and joint some occasional citations.
How could this transcriptome help the pest control, after all?

Experimental design

The experimental design was clear, but the Methods were too briefly described.
Too much abbreviations. You should always write the entire name and then the abbreviation, between parentheses, in the very first time it appears in the text. When you have so many names do abbreviate, you may even write a brief glossary.
Several tools were used, but they were weakly described. It is important always describe "why" you chose that tool, how this tool benefits your strategy. If you just cite one tool after another, the reader will not trust your methods.

Validity of the findings

There is a lot of results, and they could generate a good article, but they are immature. It is very important to have the extra information that the author describes, as transcription factors, lncRNA, but it is necessary to connect each other, show the consistency of the results as a whole.
A lot was said about the advantages of third generation sequencing strategy.
Most transcriptomes published up to now were assembled using second generation sequencing and complementary sequencing strategies to achieve the best results. Each sequencing technology has its benefits and disadvantages. The strategy is not better because a specific sequencing technology was adopted, it is better when the strategy that you adopted generated the results you were expecting. It would be advisable to call attention to the complemented long reads with the short reads, the use of short reads to adjust the high error rate of the long reads.
The full-length transcriptome described should be compared to some Arthropoda short reads assembled transcriptomes and another hybrid strategies, to show to the readers the evidence that the long reads full-length transcriptome could be advantageous, indeed.

Additional comments

Invest more in consistency. If you have some nice interesting new methodoly, describe it and show how it was important to achieve the results. In the results, show how they connect to each other. The results should tell the reader a story. The discussion must go back to the initial hypothesis, it is the moment when you stop telling the story and start talking about the repercussions, negative and positive insights, convince the reader that the content of you story was really interesting and reliable.
After all, how this results elucidated the metabolic resilience of this weevil in order to help future studies concerning red palm management and pest control?

---

## Round 0.2 · accepted · Accept

Dear Dr. Zhuo,

Congratulations for the improvements made on the revised MS.

Stay safe.

Kind regards,
Thiago

Reviewer 1 ·

Basic reporting

I reviewed an updated version of the manuscript after the first reviewing round. The manuscript was much improved by the authors, who addressed all the points raised in the first review. The authors explored more the dataset and provided all the information that was asked.

Experimental design

The authors expanded the methodology text, adding much needed details for all steps. It is now much clearer to understand and follow.

Validity of the findings

As mentioned in the previous revision, this is a descriptive paper, which aims mostly to provide the community an interesting and valid dataset, and the authors describe this novel data in a simple, straightforward way.

Additional comments

This short, straightforward, paper got much improved after the first revision round. The methods and results are much better explained, and the authors address all questions raised by the previous review. I think this manuscript provides novel data that will be much appreciate and useful to the specialists in the field.

Reviewer 2 ·

Basic reporting

The paper was properly re-written. Good language and clear sentences. Improved and consistent bibliographic references.

Experimental design

Improved and sufficiently rewritten. The experimental design was clearly described as in the fisrt version, but now it is sufficiently described.

Validity of the findings

Important results and much better exploration of them. The authors efficiently synthesized the results obtained.

Additional comments

I congratulate the authors for the excelent revision. It is clear that the authors were dedicated to improve this importante paper.